# A Novel Tool for a Challenging Disease: Stasis Leg Ulcers Assessed Using QFlow in Triggered Angiography Noncontrast Enhanced Magnetic Resonance Imaging

**DOI:** 10.3390/jpm11090857

**Published:** 2021-08-28

**Authors:** Chien-Wei Chen, Yueh-Fu Fang, Yuan-Hsi Tseng, Min-Yi Wong, Yu-Hui Lin, Yin-Chen Hsu, Bor-Shyh Lin, Yao-Kuang Huang

**Affiliations:** 1Department of Diagnostic Radiology, Chia-Yi Gung Memorial Hospital, College of Medicine, Chang Gung University, Taoyuan 333, Taiwan; chienwei33@gmail.com (C.-W.C.); ymsc10014@gmail.com (Y.-C.H.); 2Department of Thoracic Medicine, Linkou Chang Gung Memorial Hospital, College of Medicine, Chang Gung University, Taoyuan 333, Taiwan; dr.fang.yf@gmail.com; 3Division of Thoracic and Cardiovascular Surgery, Chiayi Chan Gung Memorial Hospital, College of Medicine, Chang Gung University, Taoyuan 333, Taiwan; 8802003@cgmh.org.tw (Y.-H.T.); mynyy001@gmail.com (M.-Y.W.); vw200162@gmail.com (Y.-H.L.); 4Institute of Imaging and Biomedical Photonics, National Yang Ming Chiao Tung University, Tainan 700, Taiwan; borshyhlin@gmail.com

**Keywords:** MRI, contrastless, stasis leg ulcer, ablation, surgery, QFlow

## Abstract

Imaging characteristics of stasis leg ulcers (SLUs) are not easily demonstrated through existing diagnostic tools. Early diagnosis and treatment are crucial. This pilot study was conducted to assess the quantitative flow (QFlow) in triggered angiography noncontrast enhanced (TRANCE) magnetic resonance imaging (MRI) to identify the hemodynamics of victims with stasis leg ulcers (SLUs). This study included 33 patients with SLUs and 14 healthy controls (HC). The 33 patients with SLUs were divided into a reflux (15 patients) and a nonreflux group (18 patients). QFlow was done in the reflux, the nonreflux, and the HC. The stroke volume (SV), forward flow volume (FFV), absolute flow volume (AFV), mean flow (MF), and mean velocity (MV) were higher in the reflux than in the HC group in most segments, namely the external iliac vein (EIV), popliteal vein (PV), and great saphenous vein (GSV) (SV, *p* = 0.008; FFV, *p* = 0.008; absolute stroke volume (ASV), *p* = 0.008; MF, *p* = 0.002; MV, *p* = 0.009). No differences in the QFlow patterns were found in the GSV segment between the nonreflux group and the HC. Excellent performance in discriminating SLU with superficial venous reflux was reported for SV in the EIV and the PV (area under the curve (AUC) = 0.851 and 0.872), FFV in the EIV and PV (AUC = 0.854 and 0.869), ASV in the EIV and PV (AUC = 0.848 and 0.881), and MF in the EIV and PV (AUC = 0.866 and 0.868). The cutoff levels of SV/FFV/ASV/MF in the EIV/FV/PV/GSV for discriminating the SLU with superficial venous reflux were identified (*p* < 0.005). In conclusion, SLUs present different QFlow patterns by different etiology. The QFlow parameters of all vessel segments were higher in the morbid limbs of the reflux group than HC. The GSV segment of the nonreflux group displayed a pattern like the HC.

## 1. Introduction

Stasis leg ulcers (SLUs), or venous leg ulcers, account for 80% to 90% of all leg ulcers. The wounds usually persist for more than 6 weeks and are mostly limited to the subcutaneous plane [1]. By contrast, leg wounds due to arterial diseases are rapid-progression with high risks of amputations [2,3,4]. SLUs are considered as a sequence after ambulatory venous hypertension of the legs. The cause of the venous hypertension ranged from primary venous valvular reflux, congestive heart failure, and poor lymph drainage to deep vein thrombosis (DVT). SLUs are also possibly caused by venous occlusion in the pelvic level, which is not easily demonstrated through other existed diagnostic tools (i.e., ultrasound and computed tomographic venography).

The triggered angiography noncontrast enhanced (TRANCE) technique uses differences in signal intensity of vessels during the cardiac cycle for subsequent image subtraction [5]. The morphology of the entire venous anatomy of the lower extremities, especially the low-flow superficial venous system and pelvic collaterals, can be demonstrated through three-dimensional (3D) imaging without the use of contrast medium or radiational toxicity (see video in Appendix A). TRANCE magnetic resonance imaging (MRI) is the standard preoperative evaluation modality for complex SLUs at our institution [6,7]. In this study, we analyzed venous hemodynamic parameters through quantitative flow (QFlow) in TRANCE MRI in patients with SLUs and compared them with the parameters in healthy patients to identify the patterns of complex SLUs. QFlow technique can provide phase-contrast information of the measured region of interest. Currently, QFlow technique has been used in research related to cerebrospinal fluid, aorta, and peripheral vascular disease [8,9,10,11].

## 2. Materials and Methods

### 2.1. Patients

This study has been proved by the Institutional Review Board (IRB) of Chang Gung Memorial Hospital (IRB numbers: 201802137B0, 201901058B0, and 202100938B0). The study enrolled patients who were evaluated using TRANCE MRI for venous pathologies in their lower extremities at a tertiary hospital between May 2017 and June 2021. We prospectively collected and retrospectively analyzed their QFlow to determine their clinical significance. All patients who were suspected to have complex venous pathologies in their lower extremities were initially evaluated, and only patients with SLUs and complete QFlow data were included in the study. Initially, 242 patients were evaluated, and complete QFlow analysis data were available for 168 subjects. Among these 168 subjects with complete QFlow data, 33 patients with SLUs and 10 healthy patients were included in this study. All 33 patients with SLUs underwent noninvasive color Doppler ultrasonography (US) to evaluate the venous status of their lower extremities before TRANCE MRI. The Doppler US examinations were performed in the supine position. The femoral veins (FVs), great saphenous veins (GSVs), popliteal veins (PVs), and perforating veins in the calves were examined. Pelvic veins were not checked in the US exams.

### 2.2. MRI Acquisition

MRI was done by a 1.5 T MRI scanner (Philips Ingenia, Philips Healthcare, Best, Netherlands). The image was processed with the patients in supine, and a peripheral pulse trigger was applied for imaging. All arterial system images were evaluated using a 3D turbo spin echo (TSE) skill during the systole and diastole periods. TSE TRANCE imaging was executed using the following parameters: repetition time (TR), one beat; echo time (TE), shortest; flip angle, 90°; voxel size, 1.7 × 1.7 × 3 mm^3^; field of view (FOV), 350 × 420. During systole, arterial blood flow is relatively fast, which causes signal dephasing and flow voids. Accordingly, when systolic triggering is applied, the arteries show black. During diastole, blood flow in the arteries is slow, and the signal is not dephased. Thus, the arteries appear bright on diastolic scans.

The venous system was checked through 3D TSE short-tau inversion recovery (STIR) during the systole period. TSE STIR TRANCE imaging was executed using the following parameters: TR, 1 beat; TE, 85; inversion recovery delay time, 160; voxel size, 1.7 × 1.7 × 4 mm^3^; FOV, 360 × 320. STIR gives additional background suppression because the connective tissues are also suppressed. When systolic triggering is applied, the arteries show black. The imaging process yielded a 3D dataset of the venous system, and no subtraction was required.

QFlow scans were routinely done to determine the appropriate trigger delay times for systolic and diastolic triggering. All images were acquired with no contrast media. The QFlow scan produced multiple acquisitions within one cardiac cycle (one R–R interval), resulting in several phases. The data set obtained using QFlow scanning technology contains phase shift information, which could be quantitatively analyzed by drawing a region of interest (ROI) on the primitive two-dimensional plane as an analysis. The drawing ROI was set at inferior vena cava (IVC), external iliac veins (EIVs), FVs, PVs, and GSVs for QFlow analysis (Figure 1).

After defining at least one contour area of ROI, the computer could automatically generate analysis results of various variables. These variables include stroke volume (SV), forward flow volume (FFV), backward flow volume (BFV), regurgitant fraction (RF), absolute stroke volume (ASV), mean flux (MF), stroke distance (SD), and mean velocity (MV). All these QFlow parameters were used for analysis as objective indicators.

Stroke volume (SV), mLThe net volume of blood that passes through the contour of ROI during one R–R interval.Forward flow volume (FFV), mLThe volume of blood that passes through the contour of ROI in the positive direction (toward head direction) during 1 R–R interval.Backward flow volume (BFV), mLThe volume of blood that passes through the contour of ROI in the negative direction (toward foot direction) during 1 R–R interval.Regurgitant fraction (RF), %The fraction of the backward flow to forwarding flow.Absolute stroke volume (ASV), mLThe absolute value of forwarding flow volume plus the absolute value of backward flow volume.Mean flux (MF), mL/sStroke amount × heartbeat/60 (1 R–R interval).Stroke distance (SD), cmThe net distance that blood proceeds in the vessel during 1 R–R interval.Mean velocity (MV), cm/sStroke distance × heartbeat/60 (1 R–R interval).

### 2.3. Statistical Analysis

The continuous variables (QFlow parameters) were analyzed using unpaired two-tailed Student’s t-tests or one-way ANOVA analysis of variance, and the discrete variables (sex, comorbidities, substance usage, and surgical history) were compared using two-tailed Fisher’s exact tests or Pearson’s chi-squared test. ROC analysis was performed to assess the diagnostic value of the QFlow parameters. All statistical analyses were conducted using the STATA Statistics/Data Analysis (version 8.0; Stata Corporation, College Station, TX, USA). Data are presented as means and standard deviations. Statistical significance was defined as *p* < 0.05.

## 3. Results

This study included 33 SLU patients and 14 healthy volunteers, each participant undergoing the TRANCE imaging and completing the QFlow scanning. These participants were further classified into the reflux group (15 participants with 15 diseased legs) and the nonreflux group (18 participants with 24 wounded legs) according to their TRANCE MRI/duplex scan data and compared with the healthy controls (HC group; 14 participants with 28 healthy legs). Data regarding the sex, age, comorbidities, morbid leg, CEAP (clinical, etiology, anatomy, and pathophysiology) classification, wound location, and interventions are listed in Table 1 and Table 2 [12,13,14,15,16]. Demographic data show no significant differences between the group except the age (*p* < 0.01) and the comorbidity of hypertension (*p* = 0.03). There are significant differences in the wound leg (*p* = 0.037), etiological (*p* < 0.001), anatomical (*p* < 0.001), and pathophysiology (*p* < 0.001) classifications, as well as surgical intervention (*p* = 0.001) between groups.

All 15 patients had cases of primary reflux limited to the superficial leg veins, and 10 received truncal ablation by either ARC catheters or nonthermal ablation (VenaSeal) (Figure 2).

The 18 patients in the nonreflux group are summarized in Table 1 and Table 2. Four patients with SLUs had congenital anomalies (double IVC and situs inversus), three presented with vascular compression (May–Thurner syndrome), three had SLUs related to illegal drug addiction, and six had SLUs involving both legs (33.3%). Most patients in the nonreflux group received coumadin or nonvitamin K antagonist oral anticoagulants (NOACs), and two of them underwent angioplasty for iliac vein lesions (Figure 3).

### 3.1. Comparison of Preoperative Duplex Scanning and TRANCE MRI

All 33 patients underwent preoperative duplex scanning and TRANCE MRI. The TRANCE MRI criterion of superficial venous reflux (QFlow ratio of GSV/PV >1) was checked with the duplex scan (the standard exam of the saphenous femoral junction reflux) regarding the ability to assess superficial venous reflux by a Cohen’s κ coefficient of 0.967.

### 3.2. Comparison of TRANCE MRI Hemodynamic Parameters among the Reflux, Nonreflux, and Healthy Control Groups

Table 3 shows the pairwise comparison of the QFlow parameters between groups (the reflux group, the nonreflux group, and the healthy controls). The hemodynamic parameters of TRANCE MRI, comprising SV, forward flow volume (FFV), BFV, regurgitant fraction, absolute stroke volume (ASV), MF, SD, and MV, were analyzed in the IVC, EIV, FV, PV, and GSV were compared among the reflux, nonreflux, and healthy control (HC) groups. The comparison between the QFlow of the morbid limbs in the reflux group and the HC group is summarized in Table 3. The SV, FFV, AFV, MF, and MV were higher in the reflux group than in the HC group in most segments, namely the EIV (SV, *p* = 0.008; FFV, *p* = 0.002; ASV, *p* = 0.006; MF, *p* = 0.006; SD, *p* < 0.001; MV, *p* = 0.003), FV (SV, *p* = 0.02; FFV, *p* = 0.022; ASV, *p* = 0.025; MF, *p* = 0.002), PV (SV, *p* = 0.001; FFV, *p* < 0.001; ASV, *p* = 0.001; MF, *p* = 0.004; MV, *p* = 0.013), and GSV (SV, *p* = 0.008; FFV, *p* = 0.008; ASV, *p* = 0.008; MF, *p* = 0.002; MV, *p* = 0.009). The QFlow parameters in the IVC were similar in the reflux and HC groups.

The QFlow patterns in the nonreflux and HC groups are summarized in Table 3. Only the PV segment showed significant differences in SV, FFV, ASV, MF, SD, and MF (SV, *p* = 0.002; FFV, *p* = 0.002; ASV, *p* = 0.002; MF, *p* < 0.001; SD, *p* < 0.001; MV, *p* < 0.001). A decreased MF was observed in the FV (*p* = 0.018), and a decreased SD was observed in the IVC (*p* = 0.019). Notably, no significant differences in QFlow parameters were found in the GSV segment between the nonreflux and HC groups (SV, *p* = 0.157; FFV, *p* = 0.147; ASV, *p* = 0.838; MF, *p* = 0.083; SD, *p* = 0.183; MV, *p* = 0.058). Figure 4 shows the barplots for each analyzed parameter.

The area under ROC curve (AUC) was analyzed to define an optimal cutoff value of SV, FFV, ASV, and MF, because these QFlow parameters reached a more significant differences between the reflux group and the HC group (Table 4). Excellent performance in discriminating SLU with superficial venous reflux was reported for SV in the EIV and the PV (area under the curve (AUC) = 0.851 and 0.872), FFV in the EIV and PV (AUC = 0.854 and 0.869), ASV in the EIV and PV (AUC= 0.848 and 0.881), and MF in the EIV and PV (AUC = 0.866 and 0.868) (*p*-value < 0.0001) (Figure 5). The discriminative ability between the reflux group and HC group was reported for SV in the EIV/FV/PV/GSV (*p*-values of <0.0001, 0.0034, <0.0001 and 0.0122; cutoff values of 5.23, 1.37, 0.99, and 0.33, respectively), for FFV in the EIV/FV/PV/GSV (*p*-values of <0.0001, 0.0034, <0.0001 and 0.0064; cutoff values of 5.22, 1.44, 0.99, and 0.33, respectively), for ASV in the EIV/FV/PV/GSV (*p*-values of <0.0001, 0.003, <0.0001 and 0.0082; cutoff values of 5.31, 1.36, 1.01, and 0.33, respectively), and for MF in the EIV/FV/PV/GSV (*p*-values of <0.0001, 0.0001, <0.0001 and 0.0011; cutoff values of 5.83, 1.35, 1.11, and 0.45, respectively) (Table 4).

## 4. Discussion

SLUs are skin defects that typically occur around the gaiter area (between the ankle and calf muscle) and the malleoli. Coexisting symptoms, such as itching, swelling, depression, fatigue, pain, and social isolation, are common, and the latter three problems affect up to 50% of individuals with SLUs. The main reasons for such ulcers were thought ambulatory venous hypertension from primary venous reflux, right-sided heart failure, DVT, or venous outflow obstruction [2,17,18,19]. Sometimes, other causes of unhealed leg ulcers cannot be easily differentiated through conventional venous evaluation [2,4,20]. SLUs result in a loss of productivity and present an economic burden to dressings and the healthcare system. The cost of SLU care is estimated to be $14 billion per year in the United States [21].

Most patients with SLUs undergo US at the beginning of their therapy, as the cases in this study. US is operator-dependent, provides only limited information about the pelvic area, and often obstructs the examination of wounded legs [22]. Historically, conventional venography has been the gold standard for excluding DVT in patients with SLUs. However, this is an invasive procedure that is potentially harmful to the patients. CT venography, another diagnostic tool, requires the injection of contrast media into the morbid limb to achieve optimal venous imaging of the extremities, which poses a risk to the limb [23]. Time-of-flight (TOF) MRI is less invasive than conventional venography and CT venography are and less operator dependent than US is [24,25,26]. Many MRI techniques can be used to reconstruct vascular structures, including TOF, phase contrast (PC), and electrocardiogram (ECG)-gated TSE MRI. TOF MRI became the earliest technique used to evaluate arterial pathology in 1998 [27]. However, the technique needs extraordinary time to obtain an acceptable image of the legs and is thus less clinically applicable [28,29,30]. By contrast, MRI with gadolinium-based contrast media is a more rapid manner of imaging the legs’ vasculature [31,32]. However, gadolinium-based contrast media is associated with nephrogenic sclerosing fibrosis (NSF) [33,34]. PC MRI relies on phase shifts caused by blood flow. Thus, this technique permits the use of coronal or sagittal slice orientations with an FOV along the direction of the vessel of interest and can quantitatively measure the dynamic flow of the chosen region of interest. Most prior studies have applied PC MRI to evaluate neural system pathologies, including vascular malformation and hydrocephalus [35,36].

In previous MRI techniques that do not require contrast media, such as TOF MRI and PC MRI, complete imaging of the lower extremity venous structures is highly time-intensive. The ECG-gated, multistep TSE technique (or TRANCE MRI) enables the imaging of the complete lower extremity vascular structures in clinical practice. ECG gating helps adapt imaging times to different flow characteristics and can consequently optimize image quality efficiently. Some related studies have investigated noncontrast enhanced MRI, most of which have used the technique to evaluate arterial diseases [37,38,39,40]. However, our team is the first to innovate the use of TRANCE MRI to provide more valuable morphology information; because of this, TRANCE MRI has become an important tool for managing complicated wound and venous diseases in lower extremities at our institution [6,7,41,42,43,44,45,46]. Congenital venous anomalies, May–Thurner syndrome, and pelvic congestion could be detected only by using TRANCE MRI. Coexisting peripheral arterial disease was also clearly revealed through TRANCE MRI of the arterial system. This skill is especially useful in patients with impaired kidneys, and it is valuable to avoid unnecessary compressive therapy in limbs with arterial defects [7,37]. TRANCE MRI is also effective for detecting prominent GSV reflux, an indicator for GSV truncal ablation; the interrater reliability of Cohen’s κ coefficient between TRANCE MRI and venous duplex scanning (the gold standard) was 0.967, indicating strong agreement between the two imaging modalities.

We included SLU cases with complete QFlow data and compared their patterns between the reflux group and the nonreflux group. In addition, 14 healthy volunteers (28 normal limbs) were included in an HC group for comprehensive comparisons. In the nonreflux group, the QFlow parameters were higher only in the PV segment. The GSV segment presented a QFlow pattern similar to that of the HC group. However, the QFlow analysis of the morbid limbs in the reflux group revealed more hyperdynamic circulation and fluid overload than those of the HC group; the SV, FFV, ASV, MF, and MV in all segments (including the GSV segment) were higher in the reflux group than in the HC group. The reflux group presented increased volume and flux of blood flow in the morbid limbs, including the GSV. By contrast, the QFlow parameters of the GSV in the nonreflux group were similar to those of the HC group.

In summary, this study shares our experience of using a novel application of tool (TRANCE MRI) to treat SLUs, a longstanding but challenging disease. We compared the QFlow patterns among the reflux, nonreflux, and HC groups. The QFlow parameters were higher in the morbid limbs of the reflux group than they were in the HC group. The GSV segment parameters of the nonreflux group, which were different from those of the GSV segment in the reflux group, presented a pattern similar to that of the HC group.

### Study Limitations

The major limitations of this study were its nonrandomized design and small sample size. Nonetheless, we analyzed the QFlow patterns of different etiologies (reflux vs. nonreflux) with that of healthy patients. These findings may enhance the pathogenetic discussion of SLUs.

## 5. Conclusions

TRANCE MRI is a novel appliance for treating SLUs. This study used QFlow in TRANCE MRI to analyze the hemodynamic patterns of SLUs with different causes. The QFlow parameters were higher in the morbid limbs of the reflux group compared with those of the HC group. The parameters of the GSV segment in the nonreflux group, which were different from those of the GSV segment in the reflux group, presented a pattern similar to that of the HC group.

## 6. Patents

This project is under the reviewing process in the Taiwan Intellectual property Office. (No 109126307).

## Figures and Tables

**Figure 1 jpm-11-00857-f001:**
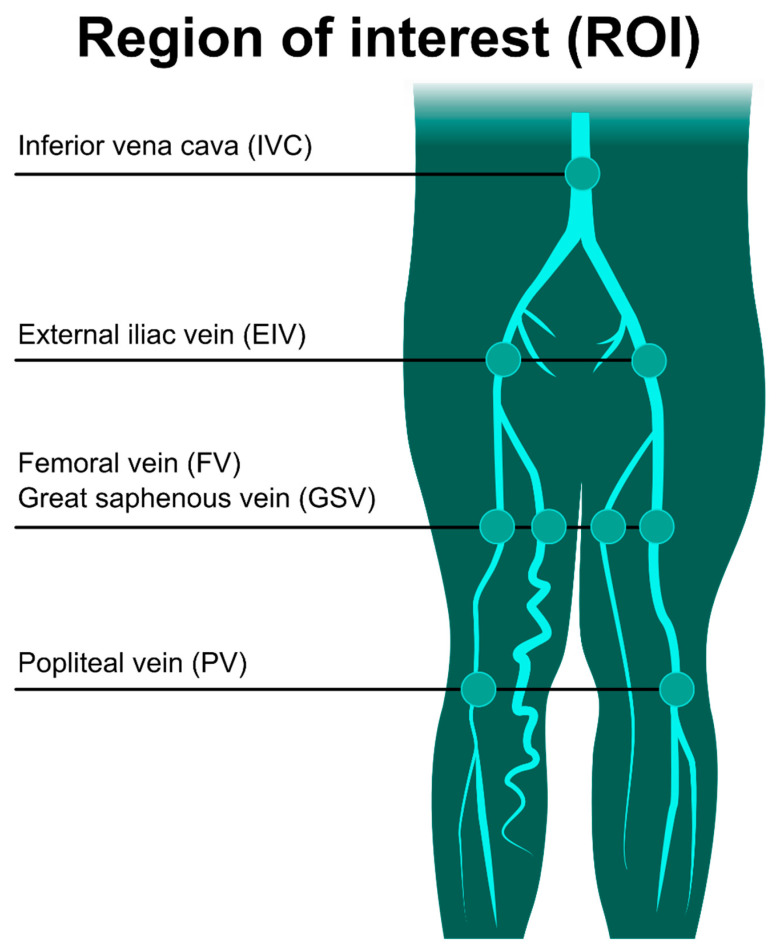
Drawing a region of interest (ROI) for the QFlow analysis. QFlow scanning is performed at four levels to obtain two-dimensional images containing phase shift information. Drawing the ROI on the vascular lumens (covering the whole lumen) obtained hemodynamic variables for the statistical analysis.

**Figure 2 jpm-11-00857-f002:**
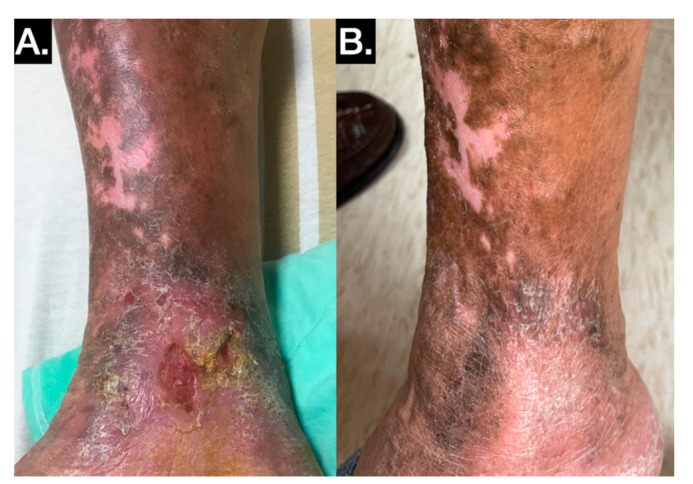
Typical stasis leg ulcer with venous reflux in the gaiter area. (**A**) Wet and nonhealing stasis leg ulcer. (**B**) Stasis leg ulcer healed after truncal ablation by ARC catheter.

**Figure 3 jpm-11-00857-f003:**
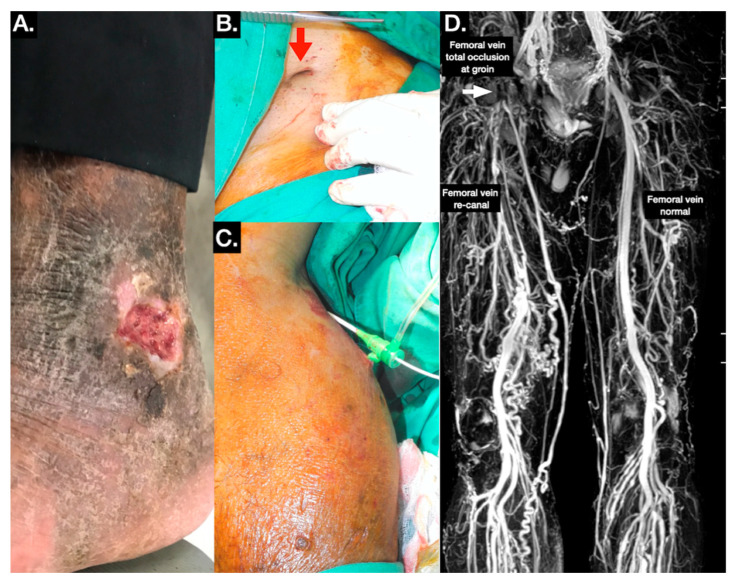
Typical triggered angiography noncontrast enhanced (TRANCE) magnetic resonance imaging (MRI) evaluation for patients with illegal-drug-use-associated stasis leg ulcers (see also video in Appendix A). (**A**) Stasis leg ulcers in the lateral gaiter area. (**B**) The punctured umbilicus in the groin indicated that this wound was related to illegal drug usage. The red arrow indicates the punctured umbilicus in the groin. (**C**) Intervention through short saphenous vein puncture combined with right internal jugular venous access. (**D**) The TRANCE MRI enabled surgeons to plan venous treatment and comprehensively explain treatments to the patients.

**Figure 4 jpm-11-00857-f004:**
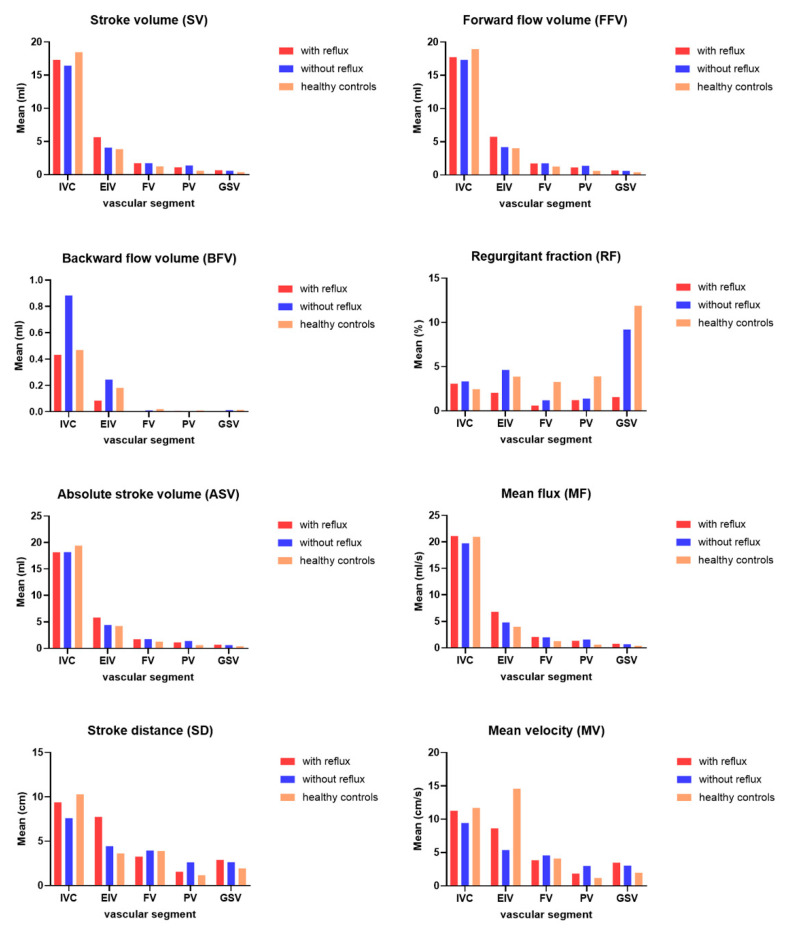
Barplots for each analyzed parameter. Each barplot presents means for each vascular segment grouped by studied group of subjects (with reflux, without reflux, and healthy controls).

**Figure 5 jpm-11-00857-f005:**
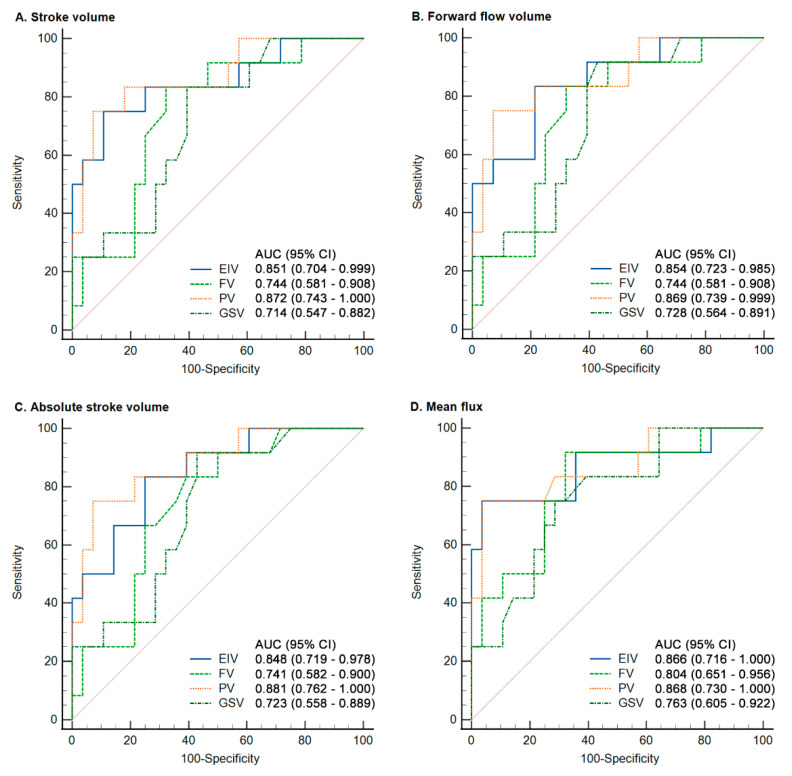
ROC curve of the SV, FFV, ASV, and MF of the QFlow analysis.

**Table 1 jpm-11-00857-t001:** Demographic characteristics of the 33 patients with SLUs.

	Total	SLU with Superficial Venous Reflux	SLU without Superficial Venous Reflux	Healthy Volunteer	*p*-Value
Patient Number	33	15	18	14	
Gender					0.262
Male	18	8	10	4	
Female	15	7	8	10	
Age	64.5 ± 11.4	59.5 ± 11.6	68.6 ± 9.7	51.4 ± 2.4	<0.01 *
Substance					
Smoking	9	3	6		0.324
Alcohol	6	1	5		0.133
Betel nut	6	3	3		0.577
Comorbidities					
Hypertension	11	2	9		0.03 *
Diabetes	9	5	4		0.373
Stroke	1	1	0		0.455
Coronary disease	0	0	0		NA
DVT history	1	9	1		0.545
Cancer	3	0	3		0.15
COPD	5	2	3		0.591
ESRD	1	0	1		0.545

DV: Tdeep vein thrombosis; COPD: chronic obstructive pulmonary disease; ESRD: end-stage kidney disease; * *p*-value < 0.05 was defined as statistically significant.

**Table 2 jpm-11-00857-t002:** CEAP classification of the 33 patients with SLUs.

	Total	SLU with Superficial Venous Reflux	SLU without Superficial Venous Reflux	*p*-Value
Patient Number	33	15	18	
Wound Leg				0.037 *
Right	16	8	8	
Left	11	7	4	
Both	6	0	6	
C in CEAP				0.455
C5	1	1	0	
C6	32	14	18	
E in CEAP				<0.001 *
Ec	4	0	4	
Ep	17	15	2	
Es	11	0	11	
En	1	0	1	
A in CEAP				<0.001 *
As	15	15	0	
Ad	13	0	13	
Ap	1	0	1	
An	4	0	4	
P in CEAP				<0.001 *
Pr	15	15	0	
Po	18	0	18	
Wound location				0.583
Gaiter area	4	1	3	
Medial ankle	20	10	10	
Lateral ankle	5	3	2	
Foot	4	1	3	
Surgical intervention				<0.001*
Conservative	11	5	6	
A.R.C catheter	8	8	0	
Venaseal	2	2	0	
NOAC	12	0	12	
Angioplasty	2	0	2	

CEAP: Clinical–Etiology–Anatomy–Pathophysiology; C5: healed venous ulcer; C6: active venous ulcer; Ec: congenital; Ep: primary; Es: secondary; En: no venous cause identified; As: superficial veins; Ap: perforating veins; Ad: deep veins; An: no venous location identified; Pr: reflux; Po: obstruction; * *p*-value < 0.05 was defined as statistically significant.

**Table 3 jpm-11-00857-t003:** Pairwise comparison of the QFlow parameters between diseased legs with reflux, without reflux, and healthy legs.

QFlow	Segments	SLU with Superficial Venous Reflux (Reflux Group, *n* = 15)	SLU without Superficial Venous Reflux (Nonreflux Group, *n* = 24)	Legs of Healthy Volunteer (HC Group, *n* = 28)	*p* Values of Pairwise Comparisons		
					Reflux and Nonreflux	Reflux and HC	Nonreflux and HC
SV	IVC	17.313 ± 6.029	16.437 ± 12.332	18.479 ± 6.518	0.823	0.599	0.475
	EIV	5.651 ± 2.182	4.078 ± 2.717	3.849 ± 1.214	0.067	0.008 *	0.705
	FV	1.738 ± 0.655	1.739 ± 1.260	1.234 ± 0.648	0.997	0.020 *	0.085
	PV	1.109 ± 0.574	1.373 ± 1.050	0.599 ± 0.316	0.380	0.001 *	0.002 *
	GSV	0.681 ± 0.410	0.592 ± 0.604	0.379 ± 0.284	0.632	0.008 *	0.157
FFV	IVC	17.747 ± 5.699	17.321 ± 13.735	18.949 ± 6.485	0.921	0.581	0.596
	EIV	5.736 ± 2.116	4.192 ± 2.859	4.033 ± 1.334	0.080	0.002 *	0.805
	FV	1.743 ± 0.646	1.750 ± 1.257	1.256 ± 0.634	0.983	0.022 *	0.090
	PV	1.117 ± 0.569	1.378 ± 1.049	0.611 ± 0.310	0.323	<0.001 *	0.002 *
	GSV	0.687 ± 0.404	0.606 ± 0.596	0.392 ± 0.272	0.661	0.008 *	0.147
BFV	IVC	0.433 ± 0.938	0.884 ± 1.792	0.469 ± 0.968	0.435	0.914	0.322
	EIV	0.085 ± 0.176	0.244 ± 0.580	0.182 ± 0.299	0.310	0.186	0.625
	FV	0.005 ± 0.018	0.010 ± 0.037	0.021 ± 0.043	0.583	0.094	0.365
	PV	0.007 ± 0.018	0.005 ± 0.021	0.010 ± 0.022	0.720	0.732	0.443
	GSV	0.005 ± 0.019	0.013 ± 0.036	0.015 ± 0.031	0.457	0.272	0.838
RF	IVC	3.083 ± 6.540	3.319 ± 5.668	2.454 ± 4.982	0.919	0.741	0.601
	EIV	2.044 ± 4.314	4.637 ± 9.739	3.887 ± 5.957	0.339	0.297	0.735
	FV	0.605 ± 2.344	1.189 ± 3.905	3.280 ± 6.963	0.605	0.073	0.181
	PV	1.203 ± 3.126	1.400 ± 5.556	3.906 ± 10.101	0.901	0.320	0.284
	GSV	1.562 ± 5.839	9.196 ± 19.809	11.890 ± 22.705	0.117	0.03 *	0.671
ASV	IVC	18.182 ± 5.508	18.206 ± 15.220	19.421 ± 6.595	0.996	0.572	0.714
	EIV	5.815 ± 2.066	4.437 ± 2.920	4.219 ± 1.505	0.120	0.006 *	0.744
	FV	1.748 ± 0.639	1.760 ± 1.255	1.279 ± 0.623	0.968	0.025 *	0.097
	PV	1.127 ± 0.563	1.383 ± 1.049	0.622 ± 0.306	0.329	<0.001 *	0.002 *
	GSV	0.693 ± 0.399	0.617 ± 0.593	0.408 ± 0.258	0.680	0.008 *	0.152
MF	IVC	21.117 ± 8.814	19.748 ± 12.251	20.986 ± 7.516	0.746	0.962	0.679
	EIV	6.814 ± 3.331	4.797 ± 3.129	3.987 ± 1.213	0.064	0.006 *	0.242
	FV	2.077 ± 0.904	1.994 ± 1.279	1.269 ± 0.682	0.829	0.002 *	0.018 *
	PV	1.346 ± 0.831	1.574 ± 1.141	0.606 ± 0.315	0.508	0.004 *	<0.001 *
	GSV	0.781 ± 0.423	0.682 ± 0.662	0.394 ± 0.310	0.625	0.002 *	0.083
SD	IVC	9.384 ± 3.808	9.384 ± 3.808	10.294 ± 4.027	0.142	0.510	0.019 *
	EIV	7.740 ± 3.046	4.439 ± 3.921	3.629 ± 0.823	0.006 *	<0.001 *	0.330
	FV	3.271 ± 1.241	3.938 ± 1.933	3.904 ± 3.068	0.242	0.345	0.964
	PV	1.551 ± 0.538	2.613 ± 1.450	1.166 ± 0.750	0.003 *	0.087	<0.001 *
	GSV	2.896 ± 1.593	2.644 ± 2.052	1.941 ± 1.552	0.702	0.070	0.183
MV	IVC	11.275 ± 5.008	9.424 ± 2.875	11.707 ± 4.477	0.228	0.789	0.074
	EIV	8.640 ± 5.092	5.381 ± 4.504	14.584 ± 56.729	0.043 *	0.689	0.432
	FV	3.836 ± 1.448	4.553 ± 2.052	4.111 ± 3.531	0.246	0.775	0.593
	PV	1.863 ± 0.792	2.996 ± 1.560	1.191 ± 0.814	0.013 *	0.013 *	<0.001 *
	GSV	3.483 ± 1.772	3.030 ± 2.137	1.982 ± 1.604	0.520	0.009 *	0.058

SLU: stasis leg ulcer; SV: stroke volume; FFV: forward flow volume; BFV: backward flow volume; RF: regurgitant fraction; ASV: absolute stroke volume; MF: mean flux; SD: stroke distance; MV: mean velocity; IVC: inferior vena cava; EIV: external iliac vein; FV: femoral vein; PV: popliteal vein; GSV: great saphenous vein; Data were presented as the mean ± standard deviation (SD). * *p*-value < 0.05 was defined as statistically significant.

**Table 4 jpm-11-00857-t004:** The AUC analysis was performed to define the optimal cutoff values of SV, FFV, ASV, and MF because these QFlow parameters reached more significant differences between the reflux group and the HC group.

Qflow	Segments	AUC	95% CI	*p*-Value	Cutoff ^#^	Sensitivity	Specificity
SV	EIV	0.851	0.704 to 0.999	<0.0001	>5.23	75	89.29
	FV	0.744	0.581 to 0.908	0.0034	>1.37	83.33	67.86
	PV	0.872	0.743 to 1.000	<0.0001	>0.99	75	92.86
	GSV	0.714	0.547 to 0.882	0.0122	>0.33	83.33	60.71
FFV	EIV	0.854	0.723 to 0.985	<0.0001	>5.22	83.33	78.57
	FV	0.744	0.581 to 0.908	0.0034	>1.44	83.33	67.86
	PV	0.869	0.739 to 0.999	<0.0001	>0.99	75	92.86
	GSV	0.728	0.564 to 0.891	0.0064	>0.33	91.67	57.14
ASV	EIV	0.848	0.719 to 0.978	<0.0001	>5.31	83.33	75
	FV	0.741	0.582 to 0.900	0.003	>1.36	83.33	60.71
	PV	0.881	0.762 to 1.000	<0.0001	>1.01	75	92.86
	GSV	0.723	0.558 to 0.889	0.0082	>0.33	91.67	57.14
MF	EIV	0.866	0.716 to 1.000	<0.0001	>5.83	75	96.43
	FV	0.804	0.651 to 0.956	0.0001	>1.35	91.67	67.86
	PV	0.868	0.730 to 1.000	<0.0001	>1.11	75	96.43
	GSV	0.763	0.605 to 0.922	0.0011	>0.45	75	71.43

AUC: area under the ROC curve; HC: healthy controls; CI: confidence interval; SV: stroke volume; FFV: forward flow volume; ASV: absolute stroke volume; MF: mean flux; EIV: external iliac vein; FV: femoral vein; PV: popliteal vein; GSV: great saphenous vein; ^#^ Determined by the Youden Index.

## Data Availability

The data presented in this study are available on request from the corresponding author. The data are not publicly available due to ethical restrictions.

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
