# Peer review of "A Novel Tool for a Challenging Disease: Stasis Leg Ulcers Assessed Using QFlow in Triggered Angiography Noncontrast Enhanced Magnetic Resonance Imaging"

_jpm, 2021, doi:10.3390/jpm11090857_

Round 1

Reviewer 1 Report

Although an interesting paper, the small number of patients and the lack of aplicability on large number of patients, reduce the paper impact.

Author Response

Reviewer 1

[Comment 1]

Although an interesting paper, the small number of patients and the lack of aplicability on large number of patients, reduce the paper impact.

[Answer 1]

Thank for your thoughtful comment. We add this weakness into “study limitations”.  Meanwhile, we revised hardly to this version, including reconstructed Figure 1, table 1, 2 and refined statistics.

Reviewer 2 Report

     The manuscript entitled „A Novel Tool for a Challenging Disease: Stasis Leg Ulcers Assessed using QFlow in Triggered Angiography Noncontrast Enhanced Magnetic Resonance Imaging” written by Chien-Wei Chen et al. presents interesting results indicating new application of TRANCE MRI examinations in diagnosis of patients with leg ulcers. Authors have revealed differences in hemodynamic parameters measured by this technique between patients with stasis leg ulcers (with or without reflux etiology) and healthy volunteers. Obtained results indicate a potential utilization of TRANCE MRI method to perform clinical assessment of patients with stasis leg ulcers with a possibility to identification of etiological factor. The reviewed work is correctly designed and scientifically significant. Overall, the text is written well and English language is appropriate and understandable. The manuscript also has some weak points and suggestions of improvements are addressed in comments below.

Broad comments:

  1. Abstract of the manuscript should be re-structured to be more adjusted to the journal requirements. Abstract should not include headings like “Background” or “Methods”. Background seems to be not addressed, because the first 5 sentences of Abstract (lines 21-25) fit rather to methodological aspects. It could be beneficial to add one or two sentences at the beginning of Abstract, to show the rationality of presented research. Moreover, some words (“reflux”, “nonreflux”, “HC”) have bolded font, is it a reason of this formatting?
  2. During reading the Introduction part, in a reader could appear a need of more information what hemodynamic parameters could be evaluated in TRANCE MRI (independently from the information about parameters used in the study) and how changes of these parameters are related to vascular diseases. Moreover, it could be add what diseases/pathological conditions/vascular segments are routinely diagnosed using this method. Providing such information will make the background of the study more complete.
  3. Statistical analysis part of Materials and Methods section include information about analyzes performed by ANOVA and Fisher’s methods, but results of these analysis are missing. Such results, regarding exploration of relationships between analyzed parameters and study group characteristics (eg. gender, comorbidities, affected venous segment, wound location) are very important for relevance of presented research.
  4. Interpretation of data included in Tables 3 and 4 could be much easier for readers by graphical presentation of this data. I suggest authors to add barplots specific for each analyzed parameter (one barplot per one parameter) and presenting means for each vascular segment grouped by studied group of subjects (with reflux, without reflux and healthy controls). Results of each comparison could be visualized on multi-panel figure containing eight barplots specific for analyzed parameter.
  5. The study include three groups of subjects: patients with reflux, patients without reflux and healthy controls, enabling three pairwise comparisons. Authors presented results regarding only patients with reflux vs. healthy controls and non-reflux patients vs. healthy controls comparisons. Adding results obtained from comparison between group of patients with reflux and group of patients without reflux could significantly enrich reviewed work, especially that presented results contribute to application of TRANCE MRI method to identification of etiological cause of leg ulcers.
  6. Presented study substantially contribute to improve diagnosis of patients with stasis leg ulcers. To deeper assess a diagnostic value of parameters obtained from TRANCE MRI and make presented results more robust and clinically relevant, data analysis could be extended with other, potentially useful methods like receiver operating characteristic (ROC) or principal component analysis (PCA).
  7. All tables should be formatted according to the journal instructions and Microsoft Word template provided by the journal.

Specific comments:

  1. Line 15 – email addresses of authors are missing
  2. Line 24 – “…with SLUs divided into…” could be changed to “…with SLUs were divided into…”
  3. Line 44 – “…which is not easily demonstrated through other existed diagnostic tools…”, please clarify other than what method you refer?
  4. Line 77 – “…image was process with…” could be changed to “…image was processed with…”
  5. In Statistical analysis part, please add information if normality of obtained data was checked.
  6. Line 112 – “…and complete QFlow analyses…” could be changed to “…and complete data of QFlow analyses…”
  7. In legend for Table 1, “\” could be removed from “…ulcer; \TPV:…”
  8. Some abbreviations seem to be not explained in the text, eg. COPD (Table 1, Table 2), ESRD, MTS, PTA, REIV, LEIV (Table 2), CT (line 192). Please check if all abbreviations are defined the first time they appear in the abstract, the main text and the first figure or table, according to the journal requirements.
  9. In tables, using more standard “None” instead of “Nil” could be more appropriate.
  10. In Tables 3 and 4 the number in “Legs of healthy volunteer” column is 28 and in line 216 the number of healthy subjects was 14, but in Abstract and Materials and Methods section, the number of 10 healthy subjects was provided. Moreover, in Table 4, column “SLU in non-reflux legs” contain number of 24, but non-reflux group include 18 subjects (lines 25 and 127). Please clarify these inconsistences.
  11. Line 186 – “…US is operator dependent, few information…” could be changed to “…US is operator dependent, provide only few information…”
  12. Line 198 – “…imaging the legs [27,28]…” could be changed to “…imaging the legs vasculature [27,28]…”
  13. Line 214-215 – “…our institution[6,7,37-42]…” could be changed to “…our institution [6,7,37-42]…”
  14. Please check if all square brackets with reference numbers are placed before the punctuation.
  15. I wonder if expression “…a novel tool…” (lines 237 and 250) in relation to TRANCE MRI is the most adequate. This method is not novel and was previously used to diagnose many others vascular diseases, as mentioned in Discussion section. Therefore, more suitable expression could be “…a novel application of tool…”

I believe that my suggestions will be helpful for authors to increase the quality of reviewed paper.

Author Response

August 25, 2021

Journal: JPM (journal of personalized medicine)

Manuscript ID: jpm-1333962

Title: A Novel Tool for a Challenging Disease: Stasis Leg Ulcers As-sessed using QFlow in Triggered Angiography Noncontrast Enhanced Magnetic Resonance Imaging.

Dear Editors, Reviewers and Ms. Lola Liu

We are submitting our manuscript entitled “A Novel Tool for a Challenging Disease: Stasis Leg Ulcers Assessed using QFlow in Triggered Angiography Noncontrast Enhanced Magnetic Resonance Imaging." for consideration of “Journal of Personalized Medicine” after revise.  Thank you very much again for granting the privilege to us to revise the paper. We have specifically responded to the reviewers’ questions and criticisms point-by-point as follows and add them into this version. Any changes in the manuscript can be tracked by the tool of the MS Word and be marked by underline.

Reviewer 2

[Comment 1]

Abstract of the manuscript should be re-structured to be more adjusted to the journal requirements. Abstract should not include headings like “Background” or “Methods”. Background seems to be not addressed, because the first 5 sentences of Abstract (lines 21-25) fit rather to methodological aspects. It could be beneficial to add one or two sentences at the beginning of Abstract, to show the rationality of presented research. Moreover, some words (“reflux”, “nonreflux”, “HC”) have bolded font, is it a reason of this formatting?  

[Answer 1]

We appreciated your informative comments for the abstract. The abstract had been restructured to meet the journal requirements[Abstract, page 1]

[Comment 2]

During reading the Introduction part, in a reader could appear a need of more information what hemodynamic parameters could be evaluated in TRANCE MRI (independently from the information about parameters used in the study) and how changes of these parameters are related to vascular diseases. Moreover, it could be add what diseases/pathological conditions/vascular segments are routinely diagnosed using this method. Providing such information will make the background of the study more complete.

[Answer 2]

Thanks for your thoughtful comments. We add descriptions in the Introduction and the Materials and Methods. Besides, we draw an illustration (Figure 1) to demonstrate how we select the regions of interest to perform the QFlow analysis.

[Introduction]

“QFlow technique can provide phase-contrast information of the measured region of interest. Currently, QFlow technique has been used in research related to cerebrospinal fluid, aorta, and peripheral vascular disease”

[Introduction] line 103-108

“The QFlow scan produced multiple acquisitions within one cardiac cycle (one R-R-interval), resulting in several phases. The data set obtained using QFlow scanning technology contains phase shift information, which could be quantitatively analyzed by drawing a region of interest (ROI) on the primitive two-dimensional plane as an analysis. The drawing ROI was set at inferior vena cava (IVC), external iliac veins (EIVs), FVs, PVs, and GSVs for QFlow analysis (Figure 1) … Stroke distance x heartbeat / 60 (1 R-R interval).”[Materials and Methods][Figure 1]

[Change for comment 2]

Change 1, line 59-61, brief introduce the clinical appliance of this technique.

Change 2: page 3 “The data set obtained using QFlow scanning technology contains phase shift information, which could be quantitatively analyzed by drawing a region of interest (ROI) on the primitive two-dimensional plane as an analysis. The drawing ROI was set at inferior vena cava (IVC), external iliac veins (EIVs), FVs, PVs, and GSVs for QFlow analysis (Figure 1).”

Change 3, line 199-223

“After defining at least one contour area of ROI, the computer could automatically generate analysis results of various variables. These variables include stroke volume (SV), forward flow volume (FFV), backward flow volume (BFV), regurgitant fraction (RF), absolute stroke volume (ASV), mean flux (MF), stroke distance (SD), and mean velocity (MV). All these QFlow parameters were used for analysis as objective indicators.

  1. Stroke volume (SV), ml

The net volume of blood that passes through the contour of ROI during one R-R-interval.

  1. Forward flow volume (FFV), ml

The volume of blood that passes through the contour of ROI in the positive direction (toward head direction) during 1 R-R-interval.

  1. Backward flow volume (BFV), ml

The volume of blood that passes through the contour of ROI in the negative di-rection (toward foot direction) during 1 R-R-interval.

  1. Regurgitant fraction (RF), %

The fraction of the backward flow to forwarding flow.

  1. Absolute stroke volume (ASV), ml

The absolute value of forwarding flow volume plus the absolute value of backward flow volume.

  1. Mean flux (MF), ml/s

Stroke amount x heartbeat / 60 (1 R-R interval).

  1. Stroke distance (SD), cm

The net distance that blood proceeds in the vessel during 1 R-R-interval.

  1. Mean velocity (MV), cm/s

Stroke distance x heartbeat / 60 (1 R-R interval).”

Change 4, new Figure one for set point

and its legends

“Figure 1: Drawing a region of interest (ROI) for the QFlow analysis. QFlow scanning is performed at four levels to obtain two-dimensional images containing phase shift information. Dawing the ROI on the vascular lumens (covering the whole lumen) obtained hemodynamic variables for the statistical analysis.”

[Comment 3]

Statistical analysis part of Materials and Methods section include information about analyzes performed by ANOVA and Fisher’s methods, but results of these analysis are missing. Such results, regarding exploration of relationships between analyzed parameters and study group characteristics (eg. gender, comorbidities, affected venous segment, wound location) are very important for relevance of presented research

[Answer 3]

Thanks for your thoughtful comments. We integrate the information of the demographic data, CEAP classification, and surgical intervention to make the new Tables (Table 1, Table 2). The new results make our research more complete and more suitable for publication.

“This study included 33 SLU patients and 14 healthy volunteers, each participant undergoing the TRANCE imaging and completing the QFlow scanning. These participants were further classified into the reflux group (15 participants with 15 diseased legs) and the nonreflux group (18 participants with 24 wounded legs) according to their TRANCE MRI/duplex scan data and compared with the healthy controls (HC group; 14 participants with 28 healthy legs).”

[Results]

“Demographic data shows no significant differences between the group except the age (p=0.02) and the comorbidity of hypertension (p=0.03). There are significant differences in the wound leg (p=0.037), etiological (p<0.001), anatomical (p<0.001), and pathophysiology (p<0.001) classifications, as well as surgical intervention (p=0.001) between groups.” [Results][Table 1 and Table 2]

[Comment 4]

Interpretation of data included in Tables 3 and 4 could be much easier for readers by graphical presentation of this data. I suggest authors to add barplots specific for each analyzed parameter (one barplot per one parameter) and presenting means for each vascular segment grouped by studied group of subjects (with reflux, without reflux and healthy controls). Results of each comparison could be visualized on multi-panel figure containing eight barplots specific for analyzed parameter.

[Answer 4]

Thanks for your suggestion.

We had drawn a multi-panel figure containing eight barplots according to this comment. Figure 4 shows the barplots for each analyzed parameter. [Figure 4](Figure 4)

[Change]

Figure legends

“Barplots for each analyzed parameter. Each barplot presents means for each vascular segment grouped by studied group of subjects (with reflux, without reflux, and healthy controls).”

[Comment 5]

The study include three groups of subjects: patients with reflux, patients without reflux and healthy controls, enabling three pairwise comparisons. Authors presented results regarding only patients with reflux vs. healthy controls and non-reflux patients vs. healthy controls comparisons. Adding results obtained from comparison between group of patients with reflux and group of patients without reflux could significantly enrich reviewed work, especially that presented results contribute to application of TRANCE MRI method to identification of etiological cause of leg ulcers.

[Answer 5]

We had added three pairwise comparisons in Table 3.    Table 3 shows the pairwise comparison of the QFlow parameters between groups ( the reflux group, the nonreflux group, and the healthy controls). [Table 3]

[Comment 6]

Presented study substantially contribute to improve diagnosis of patients with stasis leg ulcers. To deeper assess a diagnostic value of parameters obtained from TRANCE MRI and make presented results more robust and clinically relevant, data analysis could be extended with other, potentially useful methods like receiver operating characteristic (ROC) or principal component analysis (PCA).

All tables should be formatted according to the journal instructions and Microsoft Word template provided by the journal.

[Answer 6]

We performed ROC analysis to assess the diagnostic value of the QFlow parameters. We also added Table 4 and Figure 5 to explain the results.

“Excellent performance in discriminating SLU with superficial venous reflux was reported for SV in the EIV and the PV (area under the curve (AUC) = 0.851 and 0.872), FFV in the EIV and PV (AUC=0.854 and 0.869), ASV in the EIV and PV (AUC= 0.848 and 0.881), and MF in the EIV and PV (AUC=0.866 and 0.868). The cutoff levels of SV/FFV/ASV/MF in the EIV/FV/PV/GSV for discriminating the SLU with superficial venous reflux were identified (p < 0.005).” [Abstract]

“ROC analysis ws performed to assess the diagnostic value of the QFlow parameters.” [Materials and Methods]

“The area under ROC curve (AUC) was analyzed to define an optimal cutoff value of SV, FFV, ASV and MF, because these QFlow parameters reached a more significant differences between the reflux group and the HC group (Table 4). Excellent performance in discriminating SLU with superficial venous reflux was reported for SV in the EIV and the PV (area under the curve (AUC) = 0.851 and 0.872), FFV in the EIV and PV (AUC=0.854 and 0.869), ASV in the EIV and PV (AUC= 0.848 and 0.881), and MF in the EIV and PV (AUC=0.866 and 0.868) (p-value < 0.0001) (Figure 5). The discriminative ability between the reflux group and HC group was reported for SV in the EIV/FV/PV/GSV (p-values of <0.0001, 0.0034, <0.0001 and 0.0122; cutoff values of 5.23, 1.37, 0.99 and 0.33, respectively), for FFV in the EIV/FV/PV/GSV (p-values of <0.0001, 0.0034, <0.0001 and 0.0064; cutoff values of 5.22, 1.44, 0.99 and 0.33, respectively), for ASV in the EIV/FV/PV/GSV (p-values of <0.0001, 0.003, <0.0001 and 0.0082; cutoff val-ues of 5.31, 1.36, 1.01 and 0.33, respectively), and for MF in the EIV/FV/PV/GSV (p-values of <0.0001, 0.0001, <0.0001 and 0.0011; cutoff values of 5.83, 1.35, 1.11 and 0.45, respectively) (Table 4).” [Results, Figure 4, Table 4]

All the tables were reformatted according to the journal instructions

[Comment 7] Line 15 – email addresses of authors are missing

[Answer 7] We complete the author’s email: [email protected]

[Comment 8] Line 24 – “…with SLUs divided into…” could be changed to “…with SLUs were divided into…”

[Answer 8] We revise this paragraph as your correction.

[Comment 9] Line 44 – “…which is not easily demonstrated through other existed diagnostic tools…”, please clarify other than what method you refer?

[Answer 9] We add a short description here.

[Change] line 45

(ie., ultrasound and computed tomographic venography).

[Comment 10] Line 77 – “…image was process with…” could be changed to “…image was processed with…”

[Answer 10] We revise this paragraph as your correction. (line 85)

[Comment 11] In Statistical analysis part, please add information if normality of obtained data was checked.

[Answer 11] We modified the statistical analysis as your comments.

[Comment 12] Line 112 – “…and complete QFlow analyses…” could be changed to “…and complete data of QFlow analyses…”

[Answer 12] This paragraph has been re-write.

[Comment 13] In legend for Table 1, “\” could be removed from “…ulcer; \TPV:…”

[Answer 13] The table 1 has been re-built.

[Comment 14] Some abbreviations seem to be not explained in the text, eg. COPD (Table 1, Table 2), ESRD, MTS, PTA, REIV, LEIV (Table 2), CT (line 192). Please check if all abbreviations are defined the first time they appear in the abstract, the main text and the first figure or table, according to the journal requirements.

In tables, using more standard “None” instead of “Nil” could be more appropriate.

[Answer 14] The table 1 and 2 has been re-built. Thanks for your kindly help.

[Comment 15]

In Tables 3 and 4 the number in “Legs of healthy volunteer” column is 28 and in line 216 the number of healthy subjects was 14, but in Abstract and Materials and Methods section, the number of 10 healthy subjects was provided. Moreover, in Table 4, column “SLU in non-reflux legs” contain number of 24, but non-reflux group include 18 subjects (lines 25 and 127). Please clarify these inconsistences.

[Answer 15] The healthy volunteers were 14, with 28 legs. The wounds in reflux were all unilateral and the total legs were 15. The non-reflux patients were 18. Among the 18 patients, there are 6 patients have wounded on both legs. Thus the wound legs in non-reflux group were 24 in total.

We made a clarify at the beginning of the “Result.”

This study included 33 SLU patients and 14 healthy volunteers, each participant undergoing the TRANCE imaging and completing the QFlow scanning. These participants were further classified into the reflux group (15 participants with 15 diseased legs) and the nonreflux group (18 participants with 24 wounded legs) according to their TRANCE MRI/duplex scan data and compared with the healthy controls (HC group; 14 participants with 28healthy legs)

[Comment 16] Line 186 – “…US is operator dependent, few information…” could be changed to “…US is operator dependent, provide only few information…”

[Answer 16]

We revise this paragraph as your correction.

[Comment 17] Line 198 – “…imaging the legs [27,28]…” could be changed to “…imaging the legs vasculature [27,28]…”

[Answer 17] We revise the paragraph as your correction.

[Comment 18] Line 214-215 – “…our institution[6,7,37-42]…” could be changed to “…our institution [6,7,37-42]…”

Please check if all square brackets with reference numbers are placed before the punctuation.

[Answer 18] We revise all format of the footnotes as your correction in this version.

[Comment 19] I wonder if expression “…a novel tool…” (lines 237 and 250) in relation to TRANCE MRI is the most adequate. This method is not novel and was previously used to diagnose many others vascular diseases, as mentioned in Discussion section. Therefore, more suitable expression could be “…a novel application of tool…”

I believe that my suggestions will be helpful for authors to increase the quality of reviewed paper.

[Answer 19] We revise this paragraph as your correction.

For Ms. Lola Liu (Assistant Editor)

Thanks for your patience and kindly help.

Thank you for the informative and careful reviewing our manuscript. We learn lot during the revision this article. Please contact me at the following address for additional information.

Sincerely,

Yao-Kuang Huang, MD, PhD

Division of Thoracic and Cardiovascular Surgery

Chia-Yi Chang Gung Memorial Hospital, Putz, Taiwan.

Fax: 886-975368209

Round 2

Reviewer 1 Report

Altough I maintain my original opinion about the cost-efficiency and aplicability of the method, the paper was greatly improved.